# Patient and practitioner views on a combined face-to-face and digital intervention to support medication adherence in hypertension: a qualitative study within primary care

Miranda Van Emmenis [iD],[1] James Jamison,[1] Aikaterini Kassavou [iD],[1] Wendy Hardeman,[2] Felix Naughton [iD],[2] Charlotte A'Court,[1] Stephen Sutton,[1] Helen Eborall [iD][3]

[1]Public Health and Primary Care, University of Cambridge, Cambridge, UK
[2]School of Health Sciences, University of East Anglia, Norwich, UK
[3]Usher Institute, The University of Edinburgh, Edinburgh, UK

**Correspondence to**
Miranda Van Emmenis;
mv404@medschl.cam.ac.uk

## ABSTRACT

**Objectives** To explore patients' and healthcare practitioners' (HCPs) views about non-adherence to hypertension medication and potential content of a combined very brief face-to-face discussion (VBI) and digital intervention (DI).

**Methods** A qualitative study (N=31): interviews with patients with hypertension (n=6) and HCPs (n=11) and four focus groups with patients with hypertension (n=14). Participants were recruited through general practices in Eastern England and London. Topic guides explored reasons for medication non-adherence and attitudes towards a potential intervention to support adherence. Stimuli to facilitate discussion included example SMS messages and smartphone app features, including mobile sensing. Analysis was informed methodologically by the constant comparative approach and theoretically by perceptions and practicalities approach.

**Results** Participants' overarching explanations for non-adherence were non-intentional (forgetting) and intentional (concerns about side effects, reluctance to medicate). These underpinned their views on intervention components: messages that targeted forgetting medication or obtaining prescriptions were considered more useful than messages providing information on consequences of non-adherence. Tailoring the DI to the individuals' needs, regarding timing and number of messages, was considered important for user engagement. Patients wanted control over the DI and information about data use associated with any location sensing. While the DI was considered limited in its potential to address intentional non-adherence, HCPs saw the potential for a VBI in addressing this gap, if conducted in a non-judgemental manner. Incorporating a VBI into routine primary care was considered feasible, provided it complemented existing GP practice software and HCPs received sufficient training.

**Conclusions** A combined VBI-DI can potentially address intentional and non-intentional reasons for non-adherence to hypertension medication. For optimal engagement, recommendations from this work include a VBI conducted in a non-judgmental manner and focusing on non-intentional factors, followed by a DI that is easy-to-use,

### Strengths and limitations of this study

► To our knowledge, this is among the first qualitative studies to gather patient views on the use of sensing technology such as Wi-Fi or GPS within a smartphone app to support medication adherence.

► The study sought the views of a range of healthcare practitioners on incorporating a very brief intervention for medication adherence into a primary care consultation, a topic not previously explored in-depth.

► The use of stimulus materials provided detailed and focused responses for specific intervention components such as feedback on adherence and content of messages.

► While the sample size was small, the depth and focus of insights gained are sufficiently useful in informing the development and refinement of intervention components.

highly tailored and with provision of data privacy details about any sensing technology used.

## INTRODUCTION

Medication adherence, defined as the level to which an individual takes medication as intended by their healthcare prescriber, is a worldwide public health concern.[1] Non-adherence to blood pressure lowering medication is estimated at 41%, which is relatively high compared with many other medications.[2 3] This is associated with increased risk of cardiovascular disease related morbidity and mortality.[4 5] Given that high blood pressure is responsible for nearly 20% of deaths worldwide, non-adherence to antihypertensive treatment is a global health concern.[6]

Previous research into medication non-adherence has documented its complexity

and multifaceted nature.[7][8] Two broad categories within this are (1) non-intentional non-adherence, a passive process due to factors not directly within an individual's control, such as memory or access difficulties,[8][9] and (2) intentional non-adherence, a more deliberate action whereby an individual makes a conscious decision not to take their medication due to their perceptions about or experiences with their medication or condition.[9]

The multifaceted nature of non-adherence presents a challenge to those developing interventions to support adherence; for example, determining which factors to target, while balancing feasibility of delivery with likely effectiveness. Digital interventions (DIs) such as text messaging or smartphone applications (apps) offer interactive, low cost and scalable methods of providing support to individuals for whom medication adherence is a challenge. DIs are particularly suitable given the increasing use of these by people across the age groups for day-to-day tasks, such as apps for alarm clocks, calendars and shopping lists.[10] In addition, DIs can potentially lower costs compared with traditional face-to-face approaches, through reducing consultation time required with healthcare practitioners (HCPs), which may be particularly valuable at times when there is a high demand for consultations, for example, during the COVID-19 pandemic.[11][12]

Evidence for the effectiveness of DIs in improving medication adherence is promising (eg, see Thakkar *et al*[13]). In a recent systematic review of app-based interventions, patients using a smartphone app to support medication adherence for various health conditions were twice as likely to report taking their medications than those receiving usual care.[14] Specific to hypertension, DIs such as short message service (SMS) messages, smartphone apps, email and Bluetooth blood pressure monitors have been shown to improve medication adherence and lower both diastolic and systolic blood pressure.[15][16]

Incorporating sensing technology into smartphone apps potentially expands the scope of DIs further. Passive smartphone sensors can collect user location data via GPS or Wi-Fi to enable the delivery of real-time support,[17] which is of particular relevance given that non-intentional non-adherence is strongly influenced by a person's physical environment.[18] Smartphone sensing technology has shown success in DIs across the domain of health and well-being (eg, see Cornet and Holden for a review[19]) but user acceptability of such technology in a smartphone app to support medication adherence is largely unknown.

While user acceptability is key to use of a DI, potential users first need to install and engage with the DI for it to provide benefit. Primary care professionals, such as practice nurses or community pharmacists are ideally placed for encouraging uptake of DIs for medication adherence, for example, during a medication review or at the point of prescription collection. A DI used as an adjunct to a face-to-face consultation might therefore be a promising approach to support medication adherence. There is some evidence that DIs combined with tailored tele-based or web-based feedback from HCPs, improves adherence

to long-term medication[20] and antihypertensive medication.[21] However, evidence is limited on how healthcare professionals can best promote the uptake of DIs for medication adherence. The acceptability of combining a DI with a very brief face-to-face intervention (VBI) delivered by a healthcare professional to support medication adherence has also not been widely explored.

This study aimed to explore patients' and HCPs' views on (1) non-adherence to hypertension medication and (2) a complex intervention designed to support medication adherence. Initial ideas for the intervention consisted of a very brief face-to-face discussion with a primary care provider, followed by ongoing support via a DI (SMS messages or smartphone app). Feedback from participants included preferred content of the intervention and factors likely to influence engagement.

## METHODS
This study is reported in line with the Consolidated criteria for Reporting Qualitative research studies checklist (COREQ),[22] see online supplemental file 1).

### Design
We undertook a qualitative study using semi-structured interviews followed by focus groups.

### Recruitment and sampling
Patients were recruited for interviews from primary care practices based in the East of England (n=3) and East London (n=1). Practices were identified with the help of the Clinical Research Network, an organisation which supports the delivery of research within primary care in England. Patients were eligible to participate if they were: (1) prescribed at least one antihypertensive medication for at least the previous 3 months; (2) deemed non-adherent according to general practitioner (GP) practice records, with a blood pressure reading of over 140/90 mm Hg and/or gaps in filling repeat prescriptions in the previous 3 months and (3) used either SMS or smartphone apps. The practice administrator at each site generated a list of prospective participants that met criteria 1 and 2, which was screened by a GP or practice nurse. Eligible patients received a study pack from their GP practice in the post consisting of an invitation letter and participant information sheet. Posters highlighting the study were also displayed in the GP practices. Patients interested in taking part were invited to contact the researcher (MVE) via telephone or email, at which point the researcher checked that all three eligibility criteria were met before scheduling an interview.

A convenience sample of HCPs were recruited from the four GP practices taking part in this study. Healthcare practitioners were eligible to be interviewed if they were involved in the care of patients with hypertension, for example through medication reviews (conducted by a GP, practice nurse or practice pharmacist) or blood pressure checks and/or health assessments (conducted

by a healthcare assistant). The researcher invited HCPs to participate during the face-to-face study set-up meeting where they were given a study information pack. The researcher contacted the HCPs one week later to check willingness to participate and to schedule interviews for those who were interested.

Recruitment for focus groups followed that of the patient interviews. To address the low response from eligible patients, the eligibility criteria was widened to include patients prescribed medication for type 2 diabetes, as research indicates similar rates of medication non-adherence and barriers to adherence as for hypertension.[23 24] The eligibility criteria was also narrowed to ensure that participants were familiar with using smartphone apps (ie, SMS alone was not sufficient). The decision to cease individual interviews and switch to focus groups with patients was due to preliminary analysis from the interviews adding little new information to findings from previous research, and our experience of the usefulness of focus groups to gain feedback on the format, content and structure of DIs.[25–27]

### Data collection

Patient interviews were conducted by one researcher (MVE) at patients' home, workplace or local library. HCP interviews were conducted at their place of work by the same researcher (MVE). Focus groups were conducted at community centres local to the patients' general practice and moderated by two researchers (MVE and JJ).

Interviews and focus groups were guided by flexible topic guides[28] developed by the research team, drawing on the perceptions and practicalities approach (PAPA) framework[18] and previous research experience in both the topic area and intervention development.

Topic guides were reviewed by patient and public involvement (PPI) representatives to ensure the questions were easy to understand and appropriate for the study objectives. Broadly, interview topics included: reasons for medication non-adherence, current practice of HCPs during medication-related consultations, and views on a potential SMS text message or smartphone app intervention that could support adherence. Example intervention content included medication reminders, advice and support messages, and feedback on adherence. See online supplemental file 2 for the topic guides and example DI content. HCPs were shown an example protocol for a very brief face-to-face discussion or 'VBI' to generate discussion (see online supplemental file 3). Components of the VBI included: introducing the digital support to the patient and inputting basic patient information via an online questionnaire to generate the tailored digital support.

Focus groups followed similar topic guides to the interviews, focusing on attitudes towards smartphone apps in particular, including the acceptability of sensing technology such as location sensing. To prompt discussion and gain feedback, both interview and focus group participants viewed stimulus materials of example intervention messages, including medication reminders, and smartphone app features, including graphs and images (see figure 1 for examples).

Written informed consent was taken in person by the researcher immediately prior to the interviews and focus groups commencing. All patients received a £20 voucher for taking part. Interviews and focus groups were audiorecorded and professionally transcribed

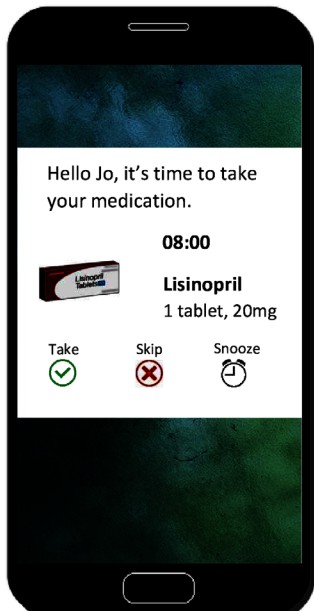
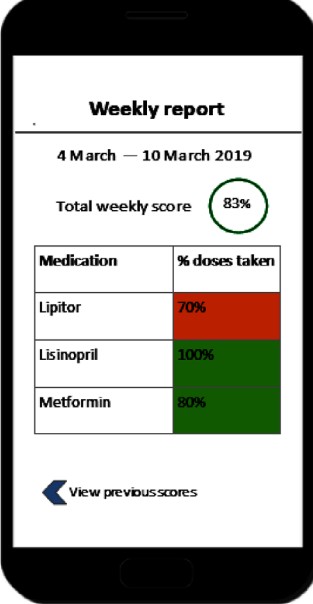
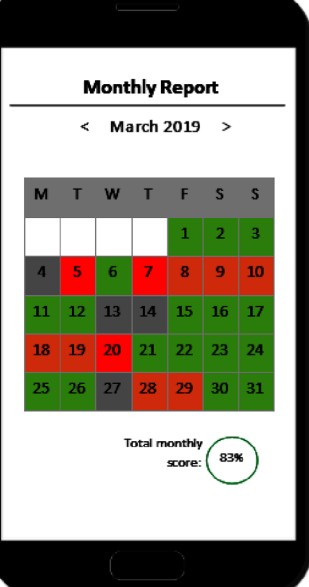

**Figure 1** . Example digital intervention content to generate discussion in interviews and focus groups; medication reminder notification, feedback on medication adherence levels (weekly and monthly), SMS support message

verbatim. Interviews lasted on average 47 minutes and focus groups 1 hour and 28 minutes.

## Data analysis

Analysis was informed methodologically by the constant comparative approach[29] and theoretically by the PAPA, which incorporates the blurring of and distinction between intentional and non-intentional non-adherence.[18] Interview transcripts were read and reread to aid familiarisation and identify preliminary themes; these broad descriptive themes were formed into an initial coding framework related to barriers and facilitators to medication adherence and a potential intervention. Each transcript was then coded systematically (MVE) using NVivo qualitative data-indexing software (V.12; QSR International) and the coding framework was refined throughout the process. The process was repeated for focus group transcripts; the coding framework was further expanded and refined, given the additional topics explored in the focus groups. A sample of interview and focus group transcripts were independently coded by a second researcher (JJ) to confirm and strengthen the validity of findings. Meetings between the research team (MVE, JJ and HE) facilitated data analysis including discussion of themes, subthemes and the interrelationships.

## Patient and public involvement

All study materials (participant information sheet, invitation letter, study poster, consent form, topic guides and stimulus materials) were reviewed by representatives from the Cambridge University Hospitals PPI panel. We made a number of changes to the study materials as a result of PPI input. We adjusted the language to make the documents more accessible and ensured interview questions were sensitively worded and easy to understand from a patient perspective. PPI representative Jennifer Bostock provided input throughout the study and reviewed and commented on this manuscript.

## RESULTS

Of the 126 eligible patients prescribed medication for hypertension who were sent an invitation, 6 were interviewed. All 11 HCPs approached by the researcher were deemed eligible and agreed to take part. Of the 218 patients prescribed medication for hypertension and/or type 2 diabetes who were then sent an invitation to a focus group, 14 participated (four focus groups with 3–5 patients per group). Recruitment of participants to focus groups continued until no new themes were emerging in relation to the specific topics covered.

Patient participant characteristics are reported in table 1. Their mean age was 62.7 years (range 47–79 years), 60% identified as male and 85% as White

| Table 1 | Participant characteristics (patients) | | |
|---|---|---|---|
| **Characteristics** | | **(n)** | **%** |
| Gender | | | |
| Female | | 8 | 40 |
| Male | | 12 | 60 |
| Age (years) | | | |
| 41–50 | | 2 | 10 |
| 51–60 | | 4 | 20 |
| 61–70 | | 11 | 55 |
| 71–80 | | 3 | 15 |
| Ethnicity | | | |
| Asian or Asian British-Indian | | 1 | 5 |
| Asian or Asian British-Pakistani | | 1 | 5 |
| Black or Black British-Caribbean | | 1 | 5 |
| White British | | 17 | 85 |
| Phone use | | | |
| SMS text messages only | | 4 | 20 |
| SMS text messages and smartphone app | | 16 | 80 |
| Data collection method | | | |
| Semistructured interview | | 6 | 30 |
| Focus group | | 14 | 70 |

n=20.
SMS, short message service.

British. Eighty per cent of patients reported using both SMS and smartphone apps, with the remaining 20% using SMS text messages only. All patients self-reported having occasionally missed or skipped their medication in the previous 3 months. HCP participant characteristics are reported in table 2; six practice nurses, two healthcare assistants, two practice pharmacists, one GP. Participants were recruited from four GP practices based in urban (n=3) and rural (n=1) locations. GP practice Index of Multiple Deprivation (IMD) scores, a measure of relative socioeconomic status in England based on postcode, ranged from 'least deprived' (n=2), to 'less deprived' (n=1) and 'more deprived' (n=1), see table 2.

To present the findings, we broadly categorise the key themes identified into the following categories: reasons for non-adherence, recommendations for message content, tailoring the DI, acceptability of sensing technology, and attitudes towards a VBI. We provide illustrative quotes below. See online supplemental file 4 for additional quotes from participants. For reference, DI refers to both SMS text messages and smartphone app, as the same intervention messages can be delivered using both formats.

### Reasons for non-adherence

Participants provided two key explanations of non-adherence to antihypertensive medication. First, for

**Table 2** Participant characteristics (healthcare practitioners) and GP practice demographics

| Characteristics | (n) | % |
|---|---|---|
| **Healthcare practitioners** | | |
| Job role | | |
| General practitioner | 1 | 9 |
| Healthcare assistant | 2 | 18 |
| Practice nurse | 6 | 55 |
| Practice pharmacist | 2 | 18 |
| Gender | | |
| Female | 10 | 91 |
| Male | 1 | 9 |
| Years practising | | |
| ≤10 | 5 | 46 |
| 11–20 | 2 | 18 |
| 21–30 | 2 | 18 |
| ≥31 | 2 | 18 |
| **GP practices** | | |
| GP practice location | | |
| Urban | 3 | 75 |
| Rural | 1 | 25 |
| GP practice IMD quintile | | |
| First quintile (least deprived) | 2 | 50 |
| Second quintile (less deprived) | 1 | 25 |
| Fourth quintile (more deprived) | 1 | 25 |

Healthcare practitioner n=11, GP practice n=4.
GP, general practitioner; IMD, Index of Multiple Deprivation, which ranks every small area in England from 'most deprived' to 'least deprived'.

non-intentional non-adherence, forgetting was the most common reported reason and was mentioned in three ways: forgetting to take medication, forgetting whether or not medication had been taken and forgetting to reorder the prescription in time.

*Sometimes you can't remember whether you have taken them already. And that can be problematic, so if someone asks you, you think, 'well, I don't know, maybe I have, maybe that was yesterday.' [P04, male, 40s]*

Second, in terms of intentional non-adherence, the experience or anticipation of side effects was a reason given for skipping, altering or delaying medication, as was the general reluctance to be reliant on medication.

*I wish I could live without medication, I hate pumping my body with drugs. Sometimes I wonder, 'what side effects am I gonna have with this? Is it really benefitting me?' [P06, female, 60s]*

*A lot of patients […have said], 'yes, the doctor has changed my medication, but they make me go funny, so I'm just going*

*to take half or I'll just crush that and just take half instead of the two.' [HCP 05, Healthcare Assistant, female]*

These overarching explanations were apparent when participants discussed the merits of a DI to improve adherence, as presented in the following sections. We begin with participants' views about a DI's messaging content, followed by tailoring and then sensing technology; the final section considers the role of the VBI component.

### Recommendations for message content

Simple reminder messages were perceived as useful for both taking medication and re-ordering prescriptions.

*Even if I'm in a hurry, [when] I receive this reminder I [would] realise the importance. I think if I keep getting messages that would be very effective and definitely help me. Even if I'm tired and it would make me […] I'd force myself to get up and go and take the medication. [P06, female, 60s]*

*It would be useful, if you're running out of tablets, to have some way of automatically reordering or a reminder to do that. So it's reminding you to take your tablets, and also when you're running low. [P04, male, 40s]*

Information-giving messages were only perceived as helpful by participants if providing advice when medication had been missed, for example the safest way to 'catch up' on a missed dose.

*There ought to be a button of 'I've forgotten them 'til now, which bits should I take?' That could be useful. [FG3, male]*

While HCP–participants recommended messages about the benefits of medication or the consequences of non-adherence, patient–participants considered these unhelpful and unnecessary, particularly if lack of knowledge was not a barrier to adherence.

*I know what the risk is [from not taking my medication]. I don't feel that I want it repeated, no. [P03, male, 60s]*

There was, however, some recognition that newly diagnosed patients may find such information motivating:

*If you're new to taking blood pressure tablets [information on consequences of non-adherence] would be good. I mean, us experienced people who've taken them for years most probably don't need reminding that if you don't take it, something serious is gonna happen to you. [FG3, male]*

The idea of receiving feedback on one's adherence, generated from self-report via SMS message or app, in a message of encouragement (eg, 'Well done!') was viewed as unnecessary. Participants were more receptive to schematic feedback in the form of a graph, score or percentage.

*Some people might need that encouragement, but then again, it sounds a bit patronising to some people, doesn't it [laughs]? […] I think the percentage thing would give people pride, you know, 'oh, I've reached 100% [of taking my medication] this month, I feel really good about that'.*

*Once a month I'd like to know what my score was for the month. I think that would probably be enough incentive for me personally. [P03, male, 60s]*

Regular smartphone users suggested that feedback in the form of a monthly calendar highlighting 'missed medication' days, could be useful for spotting a pattern and identifying the circumstances of those days that contributed to a missed dose. Moreover, participants suggested the potential for this to facilitate discussion with a healthcare professional too:

*I think [the app] would also be good to take, when you have a medication review, to take to your GP so he or she can see what's going on as well. [FG3, male]*

### Tailoring the DI

Participants commented how they would be more likely to use, and continue to use, the DI if the messages were tailored to their preferences and their individual medication regime, in terms of frequency and timing of doses:

*Some people are on medication once a day, twice a day, three, four. Could the app be tailor-made for the individual? And remind us accordingly? [FG2, female]*

*A: That's why [the intervention] should be tailor-made for the individual patient. I think it's going to be critical really. Rather than a generic –*

*B: And have options, yeah.*

*A: Because if it's a generic app and it doesn't suit some people they won't use it or they won't respond to it. [FG1, male (A,B)]*

Participants noted the importance of the DI including *all* their prescribed medications, that is, not just the hypertension ones.

*I think it would need to be somewhat of a select or deselect, you know, 'take all' but you can un-tick the ones that you're not taking now. [FG4, female]*

*A: I would do it as all one. Even if you're doing it principally motivated by blood pressure, in the sense it's, you're trying to remind us to take pills in general, aren't you, so you have to somehow-*

*B: Yes, I think you want all of them there. [FG3, male (A), female (B)]*

To ensure that tailoring meets patients' preferences and medication regime, and the changes over time, participants highlighted the importance of patients having control over the DI's settings. For example, being able to change timings of reminders and adding in short-term medication.

*A: I think I'd like to put my own [medications] in. And then when you have a "short course" [of medication] as we say, I'll add that in as well. I'd rather be in charge of putting it in.*

*B: Especially as some you have to have on an empty stomach, don't you?*

*A: Yeah, so you could fiddle with your timings for that one. [FG2, female (A), male (B)]*

*It's gotta be a dynamic thing. Medications change, dosages change, things get stopped, times may change, so I probably would see as an app which patients would be free to add and subtract. [HCP 02, GP, male]*

A 'snooze' function (similar to an alarm snooze) was well-received by participants, provided users could set their own parameters, for example, length of snooze duration and maximum number of snoozes.

*It would be good for me 'cos I'm often not home when I'm supposed to take them, so if you hit the 'snooze' for an hour or whatever you choose it to be, [...]and it'll keep reminding me again and I'll take the tablets [when I'm home]. [FG1, male]*

A suggestion for tailoring by adding images of medications into the app raised more problems than benefits; participants pointed out that *'every time you get the medication, the box changes'* [FG1] and it was felt this would create confusion, rather than help.

### Acceptability of sensing technology

Participants were initially wary about the incorporation of sensing technology, such as GPS or Wi-Fi to determine location, into an app. They raised concerns about surveillance, typically referred to as *'Big Brother'* [FG1, FG3] watching them. Participants were more likely to accept sensing technology if the perceived benefits (such as tailoring medication reminders to their specific schedule and locations) outweighed concerns about data privacy.

*It would make it impossible to forget 'cos I'd just walk through the door and take 'em. That would be brilliant. [FG1, male]*

Participants requested information to address these concerns, including: who has access to their data beyond the university (in particular, less trusted organisations such as insurance or marketing companies), where data are stored, and what happens in the event of hacking.

*A: Who are you gonna share this with? That's all I'm worried about [...]*

*B: It could be pretty valuable information for insurance companies to put their premiums up. [FG4, male (A), female (B)]*

*I think it'd be more reassuring to know it was a medical body behind it or a university body behind it; it gives it some substance and credibility. [FG1, male]*

Participants wanted to retain personal control over the sensing function, with the ability to choose when the app tracks and records their location data as well as the ability to opt in/out at any point.

*I think it would be a case of opt-in because I think some people would think it an invasion of privacy. I mean, personally*

*I think it's a good idea but, you see, some people wouldn't like it. [FG4, female]*

Discussions about sensing technology prompted participants to suggest further ideas for functions of an app. Participants in two focus groups suggested linking the sensing technology with the smartphone calendar, to proactively detect periods when away from home, triggering reminder messages to pack medication or reorder prescriptions.

*The app ought to be able to detect [that] my calendar says, 'Away for the weekend.' So the app could […] send me a message or something on the Friday to make sure I pack them. That's almost what I want to be reminded of. [FG3, male]*

Participants emphasised the need for additional features to be optional, recognising that over-complicating the DI risked disengagement from potential users.

*I suppose it's a case though of getting sufficient ability to customise it against making it just too longwinded and complicated for people to be bothered. [FG4, female]*

*I'm just trying to think of just the least steps possible for the patient, because just adding more things is going to make them less likely to use these sorts of things… It needs to just be easy for them. [HCP 01, Practice Pharmacist, female]*

Above all, participants emphasised the importance of the DI being user-friendly for the target group, many of whom may be less familiar with smartphones.

*The caveat I suppose might be that those that tend to have the chronic diseases tend to be the older age group so they may not be so tech savvy. We've got some patients who don't use mobile phones even now. [HCP 02, GP, male]*

## Attitudes towards a VBI

Patient–participants' discussions about the DI functions largely focused on addressing non-intentional non-adherence—mainly forgetting. On the whole, they were sceptical about a DI's success in addressing intentional non-adherence:

*If they're not taking the tablets and they don't wanna take the tablets, why would they sign up for the app? [FG1, male].*

HCP–participants suggested that the DI encouraged users to contact their healthcare provider if experiencing problems with their medication.

*That would be really useful in that if they're stopping it for any reason it needs to come up with a message to say, "Please make an appointment with your GP. There may be alternative medications available which would suit you and you need to make an appointment to discuss that". [HCP 03, Practice Nurse, female]*

However, a more promising way of addressing intentional non-adherence was highlighted in relation to the 5 minute VBI component prior to use of the DI. The VBI was presented as a way for HCPs to signpost patients to

the DI and discuss medication taking behaviour. HCPs talked positively about how, if done in a non-judgemental way and by an HCP with an established rapport with the patient, this could foster open communication and a more constructive consultation.

*That's the important thing, when patients can relate to you and they can see that you're actually not judging them, they do tend to then engage better. [HCP 07, Practice Nurse, female]*

A key aspect of encouraging honest communication in the VBI would be acknowledging that it is acceptable to have concerns about being prescribed medication. HCPs recommended asking the patient to talk through these concerns and, if needed, book a follow-up consultation with a prescribing practitioner about changing medication.

*Have a discussion with them as to what's been happening, what the issues are, how we can make it easier for them […] 'Is there a problem with it? Are you getting side-effects? Do you find it difficult to take?' And then we can explore some of the issues. What is really important is to sift through what the issues are. Our role in the face-to-face is actually we can explore some of these things a bit easier. [HCP 06, Practice Nurse, female]*

All HCPs perceived the VBI element as feasible to deliver within primary care and recommended incorporating it alongside a medication review or blood pressure check. HCPs had two key provisos: training to help them deliver the VBI within the tight timing of 5 minutes, and a 'user-friendly' template that could be incorporated in existing computer systems for inputting patient data to inform the subsequent DI. HCPs also noted the need for sufficient training in using the DI itself, given their role in encouraging its use in their patients following the VBI.

*I think that will be important, that whoever is talking about the app needs to know how it works and how you use it… Because if somebody who is recommending it doesn't know how to use it then you're not gonna buy into it. [HCP 09, Practice Nurse, female]*

## DISCUSSION
### Summary of main findings
Patients prescribed antihypertensive medication and the HCPs that care for them, highlighted non-intentional (forgetting) and intentional (side-effects, reluctance to medicate) reasons for their non-adherence. Participants found a mobile DI that provided simple medication reminders and feedback messages acceptable. To facilitate engagement with the DI, participants recommended it was tailored to the needs of the individual and their medication regime as well as providing user control over the tailoring and other optional functions. The use of sensing technology within a smartphone app was acceptable to participants provided they received

comprehensive information about the associated use and confidentiality of their data.

While the DI was considered limited in its potential to address intentional non-adherence, HCPs saw the potential for a brief face-to-face discussion (or 'VBI') with patients in addressing this gap, when delivered alongside a DI. Incorporating a VBI into routine primary care was considered feasible, if it could be integrated into existing practice software systems and if training were provided.

### Strengths and limitations of the study

Drawing on relevant theory,[8 9 18] this study was conducted as development work with a target patient group to inform aspects of an intervention as part of a larger research programme.[30] While previous research has investigated the use of sensing technology and smartphone apps for health,[19] this study is among the first to gather qualitative data on the acceptability of such technology (eg, Wi-Fi or GPS) in a smartphone app designed to support medication adherence (see also Kassavou et al[31]). While advances in technology can provide additional features to smartphone apps, it is important to assess the intended user group's views of such technology before its implementation.[32]

We gained insights from a range of HCPs on the acceptability and feasibility of incorporating a VBI for medication adherence into a primary care consultation, a topic that has not been previously explored in-depth. The recommendations arising from our findings can inform the development and implementation of a medication adherence VBI in primary care. Developers should consider the following: the importance of the practitioner–patient relationship when discussing medications, exploration of patient-specific barriers to adherence, templates embedded within existing GP practice systems and sufficient training for HCPs.

The use of stimulus materials generated discussion in the interviews and focus groups, and provided focused responses for specific hypothetical intervention components.

We acknowledge that this is a small-scale qualitative study, where 85% of the patient sample were White British and 91% of the HCP sample were female. As such, the findings may be limited in their application to a patient and healthcare professional population. However, the depth and focus of insights gained provided rich data that were sufficiently useful in informing the development and refinement of intervention components for the wider programme, and to similar interventions.

We experienced challenges with recruiting patients through GP practices, particularly those who were non-adherent to their medication, a group who may be less likely to participate in a study of this nature. For future studies we would recommend widening recruitment methods to include patients not tied to a specific sample of GP practices, for example, via social media channels or community groups. We acknowledge the possibility that patients who are intentionally nonadherent to their medication may be unwilling to download an adherence app or receive SMS support messages. In these instances, alternative, more intensive intervention methods involving multiple behaviour change technique components may be considered appropriate, such as motivational interviewing delivered face-to-face and/or over the telephone.[33–35]

### Comparisons with existing literature

The findings echo previous research that has identified the main reasons for non-adherence to cardiovascular-related medication as forgetting and side effects,[24 36] as well as the broad categorisation of reasons into intentional and non-intentional.[37 38] In our study, this distinction was particularly helpful when considering which elements of an intervention were appropriate for targeting these two broad categories.

Participants with lived experience of hypertension saw little value in information-style messages (eg, about the consequences of non-adherence) in addressing intentional non-adherence. Rather, they suggested that such messages may be most helpful for newly diagnosed patients. This follows previous qualitative research in which mHealth interventions were deemed especially appropriate for 'newbies',[39] that is, patients with less experience in managing a health condition compared with those with established medication routines, for atrial fibrillation[40] and type 2 diabetes.[39] Similar to previous studies,[26] participants expressed concerns about receiving too many messages, suggesting this would influence (dis)engagement with the DI. Participants also emphasised the need for a DI to be as simple and easy-to-use as possible, another common theme in usability studies for medication adherence DIs, whereby difficulties with navigating a website, SMS or smartphone app have presented barriers to usage.[41 42] A related concern is the potential burden that self-monitoring DIs place on the user, for example, asking patients to self-report their medication taking behaviour within a set timeframe.[43 44] Our findings support the need for usability testing with the DI target users, which could include assessing any associated burdens or extra responsibilities placed on the user.

Participants in this study saw the benefit that sensing technology could provide but raised data privacy concerns about its use within a medication reminder smartphone app, requesting comprehensive information and user control. Similar concerns have been identified in previous research into location-sensing apps. For individuals living with HIV, the acceptability of location-based self-monitoring reminders was dependent on the purpose of the app and who would have access to their data.[45] Similarly, young adults in Dennison[46] worried about the storage of personal location data collected by health apps and wanted control over personalising the app settings.

Despite the privacy concerns, participants in this study viewed a location-sensing smartphone app more favourably if it was created by a university or charity rather than a commercial company. This follows user feedback of

other location-based apps for smoking cessation,[47] medication adherence[31] and mental health,[48] in which apps designed by universities or for research purposes were deemed more trustworthy by participants. This reflects the discourse around the ethics of mHealth, whereby third parties and insurance companies pose potential threats to the safety of patients' health data collected by sensors or smartphone apps.[43] These ethical considerations are of particular importance given the rise of mHealth in the healthcare sphere.

## Recommendations for an intervention to support medication adherence

The findings from this study have several implications for the development of a DI to support medication adherence. To encourage engagement with an intervention, it needs to be highly tailored to each individual. This includes: the timing and content of reminder messages (to address non-intentional non-adherence) and the content of support messages (for intentional non-adherence), where knowledge and duration of health condition varies between individuals. Furthermore, a key tailoring variable as recommended by HCP–participants was the individual patient's specific barrier(s) to adherence. Tailoring data can be collected using various methods, ideally before the start of the intervention for optimal impact. This could include a short questionnaire, in person or by phone with a practitioner, within a smartphone app, or via a set of SMS messages requiring responses.[26]

It was common for participants in this study to be taking multiple medications per day, and most wanted this to be reflected in the medication reminders. This requires a balance between providing appropriate adherence support without over-complicating the DI or over-burdening the user, resulting in reduced intervention engagement.[49]

This study obtained novel insight from patients on the use of passive sensing technology within a medication adherence smartphone app. To increase the acceptability of sensing technology, future apps should explain the benefits that it can provide to the user, such as tailored medication reminders based on real-time location, or prompts to pack medication for upcoming holidays detected via calendar syncing. The app must provide a flexible opt in/out option for the collection of sensing data as well as information on how personal data will be used and stored within the app. Lastly, users may be more accepting of a location-sensing app created by a university or charity rather than a commercial company.

Primary care was viewed as an appropriate setting for HCPs to introduce patients to a DI and pair it with a brief behavioural face-to-face discussion, or 'VBI'. More specifically, this could address intentional non-adherence by exploring the specific barriers to medication adherence with patients. Using a non-judgemental approach for this, would encourage patients' openness, which in turn would provide more useful information for tailoring the DI and possibly making adjustments to the patient's regime as part of the usual care. This supports a body of literature on shared decision making,[50] which has demonstrated an association between an improved patient–professional partnership and medication adherence, for a variety of conditions[51] and for hypertension specifically.[50 52] Delivering a VBI requires skill, in order to incorporate all elements and within the short time frame.[53 54] Our findings indicate the importance of comprehensive training for healthcare professionals which incorporate the principles of shared decision making and the skills to deliver the intervention in under 5 minutes, as well as proficiency in using a DI. Lastly, the template for HCPs to complete the VBI and/or enter patients' details into the DI should be user-friendly and embedded into existing GP practice software systems.

## CONCLUSION

Overall, patients and HCPs saw the benefit of receiving medication reminders via SMS text message or smartphone app. Intervention developers should consider an intervention that is highly tailored to the user, straightforward to use, and addresses data privacy concerns. The use of sensing technology was acceptable to patients, therefore, future research could investigate the feasibility of incorporating such technology into a smartphone app for adherence. A routine primary care consultation was viewed as an appropriate setting to introduce the DI to patients and discuss medication-taking behaviour with patients, but the feasibility of delivering it as 'very brief' that is, under 5 minutes, should be explored further.

Online supplemental material S1—COREQ checklist S2—Topic guides and sample of proposed intervention content S3—Example VBI protocol S4—Extra participant quotations

**Acknowledgements** This study was conducted on behalf of the Programme on Adherence to Medication team (see https://www.phpc.cam.ac.uk/pcu/research/ research-projects-list/other-projects/pam/ for team members). We acknowledge and thank the following individuals for contributing to the development of the topic guides and visual prompts for this study: Jagmohan Chauhan and Sandra Servia, Department of Computer Science and Technology, University of Cambridge; Debi Bhattacharya, School of Pharmacy, University of East Anglia.We thank the patients, healthcare practitioners, practice managers and administrative staff at the 4 GP practices who took part in this study. We are grateful to Patient and Public Involvement (PPI) representatives for reviewing the design and ethics of this study. We thank PPI representative Jennifer Bostock for their input across the whole study and for reviewing the manuscript. We acknowledge the support of the National Institute for Health Research Clinical Research Network (NIHR CRN).

**Contributors** All authors made substantial contributions to the conception and design of the study. MVE conducted the interviews, cofacilitated the focus groups, conducted data analysis and drafted the manuscript. JJ cofacilitated the focus groups, contributed to data analysis and drafted the manuscript. HE supervised data collection, contributed to data analysis and drafted the manuscript. SS, FN, WH, AK and CA critically revised the manuscript and provided intellectual input and expert advice. All authors have read and approved this manuscript. MVE is responsible for the overall content as guarantor.

**Funding** This article presents independent research funded by the National Institute for Health Research (NIHR) under the Programme Grants for Applied Research programme (RP-PG-0615-20013).

**Competing interests** None declared.

**Patient consent for publication** Not applicable.

**Ethics approval** Ethical approval for this study was obtained from the West Midlands – Solihull Research Ethics Committee, as part of NHS Health Research Authority approvals (Reference: 18/WM/0050).

**Provenance and peer review** Not commissioned; externally peer reviewed.

**Data availability statement** Data are available on reasonable request. Data are available on reasonable request from the corresponding author.

**ORCID iDs**
Miranda Van Emmenis http://orcid.org/0000-0002-4717-6746
Aikaterini Kassavou http://orcid.org/0000-0002-6562-4143
Felix Naughton http://orcid.org/0000-0001-9790-2796
Helen Eborall http://orcid.org/0000-0002-6023-3661

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
