## [Reviewer comments · BMJ Open]

ARTICLE DETAILS

TITLE (PROVISIONAL)	Patient and practitioner views on a combined face-to-face and digital intervention to support medication adherence in hypertension: a qualitative study within primary care
AUTHORS	Van Emmenis, Miranda; JAMISON, JAMES; Kassavou, Aikaterini; Hardeman, Wendy; Naughton, Felix; A'Court, Charlotte; Sutton, Stephen; Eborall, Helen

VERSION 1 – REVIEW

REVIEWER	Jongsma, Karin University Medical Center Utrecht
REVIEW RETURNED	19-May-2021

GENERAL COMMENTS	The authors present a qualitative study into the perspectives of (potential) end users of a potential intervention to enhance therapy adherence for hypertension. The study is well-described and the rationale for supplementing patient interviews with focus groups is clearly described and reasonable. My main concern with the text is that as far as I can see, the question as to whether such a techno-fix (app/or text message service) is at all desirable and needed to enhance therapy adherence. The findings, while overall interesting (but also not very surprising), seem to mostly direct to guidance, reminders and the type of information patients need, which –again- as far as I can see, does not demand a digital solution. Clarification as to whether better training of physicians or nurses (potentially supplemented with a simple alarm clock for reminders) wouldn't be an equally good or even better solution would be necessary to improve the discussion and conclusion. This is especially prudent, given that – as the authors outline- a digital solution would only help those that unintentionally do not adhere to their medication, whereas those that intentionally are not adherent would not be helped with such a digital solution. Additionally, could the authors (in the discussion) reflect on the relation between sensing technology for therapy adherence and related ethical issues such as: the domestication of patients via these technologies, responsabilisation and the burden of invisible work' of patients. See amongst others: Lupton D. (2013) The digitally engaged patient: self-monitoring and self-care in the digital health era. Social Theory & Health; 11:256-70.
---

	Lucivero F, Jongsma KR (2018). A mobile revolution for healthcare? Setting the agenda for Bioethics. Journal of Medical Ethics. 44:685–689 Oudshoorn N. Diagnosis at a distance: the invisible work of patients and healthcare professionals in cardiac telemonitoring technology. Sociol Health Illn 2008;30:272–88. Some minor aspects:  - A quote of a HCP is used to illustrate why patient do not take their medication. To what extend are they able to voice this for patients? - Please record how many focus groups have been conducted in the abstract and methods section
--	---

REVIEWER	Ferdinand, K Division of Cardiology, Tulane University School of Medicine, 1430 Tulane Avenue, SL-48, New Orleans, LA, 70112, USA
REVIEW RETURNED	19-Jun-2021

GENERAL COMMENTS	It would be best to not attempt to ameliorate the limited sample size by saying “relatively”. The paper should clarify role of researcher or research team since it is implied that the researcher directly contacted all the participants to see if they were interested. Is this correct? Also, were the interviews done by the healthcare practitioner the same person as the researcher? The social determinants of health have a profound impact on many aspects of care, including adherence. Did the researchers have any markers for social economic status, neighborhood and/or education that may impact patient’s adherence? If possible, can the authors of the paper should consider removing the personal commentaries. These vignettes appear to lengthen the text without informing the reader as to the validity of the study or the usefulness of the study. The researchers should state how they determined patients perception of the value of the tool. For example, did they collect a Likert Scale, which would facilitate some attempt to quantify patients views. Lines in the text to review for possible minor changes: 587-590. It still remains unclear to what extend the responses to the subjects of the value of the intervention was quantified or if there was any attempt to quantify. It is unclear to us whether this should be called a study or a report. The small number and collection of short personal vignettes with no apparent effort to quantify the participants opinions suggest that this is more of a "report" than a "study".
--

REVIEWER	Mostarac, Ivona Sunnybrook Health Sciences Centre
REVIEW RETURNED	12-Aug-2021

GENERAL COMMENTS	Thank you for the opportunity to review your manuscript. This qualitative study explores the patient and healthcare provider perspective on noncompliance with antihypertensive medications and the use of digital interventions and face-to-face discussions to improve medication compliance. Introduction
--

	Lines 91-95: You report previous research here about medication non-adherence having two broad categories of non-intentional non-adherence and intentional non-adherence. This is also mentioned in detail in your results. Were the findings from the previous research used for deductive coding in your study? If so, you need to report this in your methods. Methods Lines 141-142: I am not familiar with the "Clinical Research Network" and other international readers may not be as well. I would provide a little bit of information about this organization. Lines 143-147: How many patients were deemed eligible (criteria #1 and #2) and were contacted to participate? How many were not included because they did not meet criteria #3? I think this is could be important information and should be mentioned in the results section as not everyone has access to a smart phone or the knowledge/capability to use one. Lines 160-161: How many healthcare practitioners were eligible to participate? How many wanted to participate? I would include this in the results section. Lines 162-169: You report the rationale for adding focus groups but do not explicitly state why you have added patients with type 2 diabetes to the eligibility criteria. I would assume this is due to low uptake of hypertensive patients but I think you need to explain why you chose patients with type 2 diabetes as opposed to another co-morbidity. Do patients with type 2 diabetes experience the same barriers to medication adherence as hypertensive patients? Additionally, you widened recruitment to also include type 2 diabetic patients but then narrowed the criteria to include those familiar with smart phone apps as opposed to just SMS messaging. Why is that? Lines 171-175 and supplementary file #1: More information needs to be provided on the researchers' credentials, specifically looking at the training provided to conduct qualitative research. Results Lines 233-235: Were there any differences in themes between hypertension, type 2 diabetes and combo hypertension/type 2 diabetes focus groups? Line 237: 85% of patient participants were White British. This needs to be added as a limitation. Line 241: What is the role of a healthcare assistant? A brief description might be beneficial to international readers. Table 2: 91% of healthcare participants were female, this should be listed as a limitation. Lines 288-290: How would an app or SMS text messaging help the individuals who forget if they have taken their medications? How is this different or better than someone using a dated pharmacy organized blister pack or a patient organized week long pill dispenser where missed doses can be easily identified?
--	--

	Data saturation is mentioned in the supplementary file #1 but is not mentioned in the methods or the result section.
--	--

VERSION 1 – AUTHOR RESPONSE

Reviewer: 1

Dr. Karin Jongsma, University Medical Center Utrecht Comments to the Author:

The authors present a qualitative study into the perspectives of (potential) end users of a potential intervention to enhance therapy adherence for hypertension. The study is well-described and the rationale for supplementing patient interviews with focus groups is clearly described and reasonable.

- My main concern with the text is that as far as I can see, the question as to whether such a techno-fix (app/or text message service) is at all desirable and needed to enhance therapy adherence.
- The findings, while overall interesting (but also not very surprising), seem to mostly direct to guidance, reminders and the type of information patients need, which –again- as far as I can see, does not demand a digital solution.
- Clarification as to whether better training of physicians or nurses (potentially supplemented with a simple alarm clock for reminders) wouldn't be an equally good or even better solution would be necessary to improve the discussion and conclusion. This is especially prudent, given that –as the authors outline- a digital solution would only help those that unintentionally do not adhere to their medication, whereas those that intentionally are not adherent would not be helped with such a digital solution.

Response: Thank you for these useful points. On the whole, our participants were positive about the potential of the digital element to address the more non-intentional side of non-adherence, if tailored accordingly. While there are alternatives to digital solutions, the growing use of smartphones for other day-to-day tasks (e.g. as an alarm clock, timer, calendar, shopping list, etc.) – including by older age groups – presents an easy 'tool' through which an intervention (or part of a complex intervention) can be delivered at relatively low cost.

DIs offer benefits over more traditional face-to-face approaches due to its potential to be scaled up at relatively low cost, and it does not require lengthy consultations with practice nurses, whose time is scarce, particularly during the Covid pandemic. We have added some sentences to the introduction to emphasise the benefits of DIs over face-to-face interventions to support medication adherence (please see line 105 onwards).

This work did not aim to identify all such potential approaches, but to investigate views on this particular approach. The likely solution to any health problem with a behavioural component is a multi-level approach with more than one single intervention. The evidence of effectiveness of DIs across various health conditions is promising. This small study was part of a larger research programme to develop and test a combined face-to-face and digital intervention; hence, we were keen to find out whether, and how best the digital element could play a role, in order to inform intervention development.

Additionally, could the authors (in the discussion) reflect on the relation between sensing technology for therapy adherence and related ethical issues such as: the domestication of patients via these technologies, responsabilisation and the burden of invisible work' of patients.

See amongst others:

Lupton D. (2013) The digitally engaged patient: self-monitoring and self-care in the digital health era. *Social Theory & Health*; 11:256-70.

Lucivero F, Jongsma KR (2018). A mobile revolution for healthcare? Setting the agenda for Bioethics. *Journal of Medical Ethics*. 44:685–689

Oudshoorn N. Diagnosis at a distance: the invisible work of patients and healthcare professionals in cardiac telemonitoring technology. *Sociol Health Illn* 2008;30:272–88.

Response: We have now added a section in the discussion outlining some of the potential risks that DIs present for patients, including the burden of self-monitoring and the safety of their health data. (Please see lines 610 - 623).

Some minor aspects:

- A quote of a HCP is used to illustrate why patient do not take their medication. To what extend are they able to voice this for patients?

Response: This is a valid point. The paper includes this quote because we wanted to hear what patients tell HCPs about medication (non)adherence during consultations, from the HCP perspective. It also demonstrates how some patients alter their doses, rather than simply skipping the dose entirely. For these reasons, we have respectfully decided to keep the quote in the manuscript.

- Please record how many focus groups have been conducted in the abstract and methods section

Response: This information has now been added to the Abstract. Please also see the results section, line 259.

Reviewer: 2

Dr. K Ferdinand, Division of Cardiology, Tulane University School of Medicine, 1430 Tulane Avenue, SL-48, New Orleans, LA, 70112, USA

Comments to the Author:

It would be best to not attempt to ameliorate the limited sample size by saying “relatively”.

Response: We have deleted the word ‘relatively’ to accurately reflect the sample size limitation (lines 80 and 582).

The paper should clarify role of researcher or research team since it is implied that the researcher directly contacted all the participants to see if they were interested. Is this correct?

Response: Patients received an invitation letter from their GP practice instructing them to contact the researcher if interested in taking part. Healthcare practitioners met the researcher face-to-face and were given the study information sheet to take away. The researcher then contacted HCPs within a week to confirm willingness and availability to schedule an interview.

The ‘recruitment and sampling’ section has been amended to clarify the recruitment process (lines 162 and 173 – 177)

Also, were the interviews done by the healthcare practitioner the same person as the researcher?

Response: The same researcher (MVE) interviewed both patients and healthcare practitioners. This information has been added to the manuscript (line 193).

The social determinants of health have a profound impact on many aspects of care, including adherence. Did the researchers have any markers for social economic status, neighborhood and/or education that may impact patient's adherence?

Response: Please see Table 2 for rural/urban indicators for the four GP practice locations and Index of Multiple Deprivation scores, based on GP postcodes in England. These provide a crude measure of socio-economic status of the participants registered at these practices. We have added a sentence summarising these characteristics to the Results section (lines 268 – 272).

If possible, can the authors of the paper should consider removing the personal commentaries. These vignettes appear to lengthen the text without informing the reader as to the validity of the study or the usefulness of the study.

Response: Including excerpts of interviews and focus groups is standard practice (and usually a requirement) in qualitative research in order to provide evidence for the arguments and statements made in the text.

- The researchers should state how they determined patients perception of the value of the tool. For example, did they collect a Likert Scale, which would facilitate some attempt to quantify patients views.
- Lines in the text to review for possible minor changes: 587-590. It still remains unclear to what extend the responses to the subjects of the value of the intervention was quantified or if there was any attempt to quantify.

Response: We will address these comments together. Thank you for your suggestions; we agree that adding a quantitative component would provide useful additional data for a larger mixed-methods study. Indeed, other sub-studies of this larger research programme are quantitative in nature. However, as detailed in the methods section, this study was purely qualitative in nature and drew upon the principles of an interpretivist qualitative paradigm, not a quantitative one. We believe a qualitative study was justified due to the exploratory nature of the study aims; before developing an intervention, we must first assess the patients' barriers to adherence and acceptability of the hypothetical intervention. Our exploration of patients' perceived value of the digital intervention was done in depth, following systematic analysis outlined in the methods section, as typical for a qualitative study. As with most qualitative studies, our study did not seek to quantify the data.

It is unclear to us whether this should be called a study or a report. The small number and collection of short personal vignettes with no apparent effort to quantify the participants opinions suggest that this is more of a "report" than a "study".

Response: Please see the COREQ statement (supplementary file 1) which indicates that this study fulfils the requirements of a qualitative research study.

Reviewer: 3

Ms. Ivona Mostarac, Sunnybrook Health Sciences Centre

Comments to the Author:

Thank you for the opportunity to review your manuscript. This qualitative study explores the patient and healthcare provider perspective on noncompliance with antihypertensive medications and the use of digital interventions and face-to-face discussions to improve medication compliance.

Introduction

Lines 91-95: You report previous research here about medication non-adherence having two broad categories of non-intentional non-adherence and intentional non-adherence. This is also mentioned in detail in your results. Were the findings from the previous research used for deductive coding in your study? If so, you need to report this in your methods.

Response: The Perceptions and Practicalities Approach (PAPA) incorporates and acknowledges the blurring of and distinction between intentional and unintentional non-adherence in its model. Our analytical approach was informed methodologically by the constant comparative approach and theoretically by PAPA, so it was largely inductive in the early stage and interpretative, including drawing on PAPA, in the later stages. We have now added some text about this to the methods section to highlight PAPA's inclusion on intentional/unintentional non-adherence (line 226).

Methods

Lines 141-142: I am not familiar with the "Clinical Research Network" and other international readers may not be as well. I would provide a little bit of information about this organization.

Response: A sentence has been added which explains the role of the Clinical Research Network (lines 153 – 155).

Lines 143-147: How many patients were deemed eligible (criteria #1 and #2) and were contacted to participate? How many were not included because they did not meet criteria #3? I think this is could be important information and should be mentioned in the results section as not everyone has access to a smart phone or the knowledge/capability to use one.

Response: The Results section now includes information on the number of patients invited for interviews and focus groups (line 255 – 260).

Regarding criteria #3, please see the Results section (line 264-265), and Table 1 which shows that 20% of participants didn't use smartphone apps (these four participants were from the one-to-one interviews). All participants who contacted the researcher about the focus groups had a smartphone, because this requirement was listed on the information sheet/poster.

Lines 160-161: How many healthcare practitioners were eligible to participate? How many wanted to participate? I would include this in the Results section.

Response: This information has now been added. Please see the Results section (line 256)

Lines 162-169: You report the rationale for adding focus groups but do not explicitly state why you have added patients with type 2 diabetes to the eligibility criteria. I would assume this is due to low uptake of hypertensive patients but I think you need to explain why you chose patients with type 2 diabetes as opposed to another co-morbidity. Do patients with type 2 diabetes experience the same barriers to medication adherence as hypertensive patients? Additionally, you widened recruitment to also include type 2 diabetic patients but then narrowed the criteria to include those familiar with smart phone apps as opposed to just SMS messaging. Why is that?

Response: Thank you for pointing this out; we have amended the methods section accordingly. We widened the criteria due to a low response rate and because research indicates similar rates of

medication nonadherence and barriers to adherence for people with hypertension and type 2 diabetes. We narrowed to smartphone users to make sure that the data was focused. Amended text begins on line 178.

Lines 171-175 and supplementary file #1: More information needs to be provided on the researchers' credentials, specifically looking at the training provided to conduct qualitative research.

Response: Details of qualitative training (MVE) has been added to supplementary file #1 – COREQ checklist.

Results

Lines 233-235: Were there any differences in themes between hypertension, type 2 diabetes and combo hypertension/type 2 diabetes focus groups?

Response: No difference in themes were identified between participants related to type of prescription. We have not included this in the manuscript – due to word count constraints, but could add it in if requested by the Editor.

Line 237: 85% of patient participants were White British. This needs to be added as a limitation.

Response: We have added this as a limitation. Please see Strengths and Limitations section (line 580).

Line 241: What is the role of a healthcare assistant? A brief description might be beneficial to international readers.

Response: The Methods section has been updated to indicate the role that healthcare assistants have in supporting patient adherence (line 171).

Table 2: 91% of healthcare participants were female, this should be listed as a limitation.

Response: We have added this as a limitation. See Strengths and Limitations section, line 581.

Lines 288-290: How would an app or SMS text messaging help the individuals who forget if they have taken their medications? How is this different or better than someone using a dated pharmacy organized blister pack or a patient organized week long pill dispenser where missed doses can be easily identified?

Response: Pill dispensers can certainly help patients to identify missed doses, but if the patient struggles to take their medication they may also forget to use the dispenser. There is evidence that DIs, such as text messages and smartphone apps, help people adhere to their medications for a range of chronic health conditions, when compared to usual care (e.g. see Thakkar et al. 2016 JAMA Intern Med and Armitage et al. 2020 BMJ Open). The DI proposed in this paper aims to aid adherence with a message/alarm at the exact time that the medication dose is due, and encourage the user to engage with the technology by reporting if they have taken or missed a tablet. This is in contrast to a pillbox which acts as a passive reminder, showing that the medication has or has not been taken.

Participants raised another benefit that smartphone apps can have, in the form of a weekly or monthly calendar highlighting which days the user missed their medication (the behaviour change technique 'self-monitoring'), which could be useful for spotting a pattern (please see lines 380 - 390).

The proposed DI could be used as an adjunct to other interventions such as a pill dispenser.

Data saturation is mentioned in the supplementary file #1 but is not mentioned in the methods or the result section.

Response: We would argue that claiming full data saturation from a small qualitative study is highly unlikely and many authors' claims to saturation could be questioned (e.g. see Braun & Clarke 2021 for a discussion on the use of the term data saturation doi.org/10.1080/2159676X.2019.1704846) We prefer instead to argue that no new themes were emerging in relation to the specific focus covered by our topic guides. A sentence summarising this has been added to the results section (please see line 260) and the COREQ checklist (supplementary file 1, page 4).

Additional changes made to manuscript submission

Supplementary files

- COREQ checklist – page numbers have been updated to reflect the revised 'clean' version of the manuscript

Manuscript

- The word count has been updated (line 23)

- Abbreviations: added IMD, NHS and NIHR CRN (line 727)

VERSION 2 – REVIEW

REVIEWER	Jongsma, Karin University Medical Center Utrecht
REVIEW RETURNED	05-Oct-2021

GENERAL COMMENTS	Thank you for the opportunity to read the revised version of the manuscript. The authors have addressed most of the specific points from the previous review and I believe the manuscript has improved by the adjustments. Yet, there are some remaining issues in line with my previously pointed out remarks: The authors seem to be convinced of the need of DI to improve adherence, while there are also other alternatives available, especially if the demanded function of such apps is a reminder as your participants outline. Reservation as to whether such an DI is the best solution for the issue at hand would be appropriate (especially as the authors do not make a comparison with other measures to improve adherence), and should also be reflected in the tone of the paper: [ ] The authors write that “In addition, the benefit of DIs over traditional face-to-face approaches is that they do not require lengthy consultations with healthcare practitioners, whose time is scarce, especially during the current Covid pandemic.” There is no reference to support this claim here (that Di can replace face to face consultation), and while I see the issue of time being scarce for health practitioners, this phrasing seems somewhat exaggerated. If an adherence conversation would indeed demand a lengthy consultation, would it then be easily resolved by an app? And if so, would that be desirable (for whom)? I think the authors intent to make a more general claim here, either about cost-effectiveness or efficiency, please consider rephrasing [ ] Discussion: Given the low recruitment/willingness to participate, I am somewhat surprised that the authors do not discuss this in their paper: f only so few people are willing to use the app when its readily available? [ ] Discussion, the authors write: ‘Participants felt a mobile digital intervention (DI) that provided simple medication reminders and feedback messages would help improve adherence.’ I’m not sure
--

	this follows from your results, your participants argue it would be helpful, but not that it would improve adherence (as you outline yourself, adherence is more complex than simply forgetting a dose) □ Discussion: 'and is among the first qualitative studies to gather patient views on the use of sensing technology (e.g. Wi-Fi or GPS) in a smartphone app to support medication adherence (see also Kassavou and colleagues[29]).' There are several studies and reviews about adherence apps -also some qualitative studies- that aren't referred to and aren't used to contextualize and juxtapose the findings of this study. I would highly encourage the authors to carefully review the existing evidence. eg: https://mhealth.jmir.org/2019/1/e11919/ , https://pubmed.ncbi.nlm.nih.gov/27248315/ □ Discussion, the authors write 'We gained novel insights from a range of healthcare practitioners on the acceptability and feasibility of incorporating a VBI for medication adherence into a primary care consultation, a topic that has not been previously explored in-depth.' Could the authors outline here what the new findings are? □ Discussion, the authors write 'Our findings can inform the development and implementation of a medication adherence VBI in primary care.' Please outline how?
--	--

REVIEWER	Mostarac, Ivona Sunnybrook Health Sciences Centre
REVIEW RETURNED	21-Oct-2021

GENERAL COMMENTS	Thank you for the opportunity to review your revised manuscript. I appreciate your consideration of my feedback and I feel you have addressed my concerns adequately in your revision.
--

VERSION 2 – AUTHOR RESPONSE

Reviewer: 1

Dr. Karin Jongsma, University Medical Center Utrecht Comments to the Author:

Thank you for the opportunity to read the revised version of the manuscript. The authors have addressed most of the specific points from the previous review and I believe the manuscript has improved by the adjustments.

Yet, there are some remaining issues in line with my previously pointed out remarks: The authors seem to be convinced of the need of DI to improve adherence, while there are also other alternatives available, especially if the demanded function of such apps is a reminder as your participants outline. Reservation as to whether such an DI is the best solution for the issue at hand would be appropriate (especially as the authors do not make a comparison with other measures to improve adherence), and should also be reflected in the tone of the paper:

Response:

- The intervention proposed in this study is a combination of both digital and face-to-face support. The digital element (SMS messages or smartphone app) is able to deliver tailored support to users, including simple reminders and tailored feedback, as and when needed, in addition to any healthcare consultations. We are proposing that a DI delivered alongside a brief face-to-face intervention with healthcare staff could be acceptable to patients.
- There is a drive for low-cost digital solutions in healthcare e.g. NHS 'digital first' practices (<https://www.longtermplan.nhs.uk/online-version/chapter-1-a-new-service-model-for-the-21st-century/4-digitally-enabled-primary-and-outpatient-care-will-go-mainstream-across-the-nhs/>) and WHO guidance on integrating digital interventions into existing healthcare systems (<https://www.ncbi.nlm.nih.gov/books/NBK541905/>). This coupled with evidence for the effectiveness of DIs in supporting medication adherence (e.g. please see references 13-16) provides rationale for investigating the acceptability of digital interventions.
- We have made a number of amendments in the manuscript in response to the reviewer's recommendations.
- We have added a sentence in the introduction to clarify that the DI proposed in this paper is designed to be delivered alongside a F2F appointment instead of replacing traditional existing primary care consultations.
 - o Line 131 "A DI used as an adjunct to a face-to-face consultation might therefore be a promising approach to support medication adherence. There is some evidence that DIs combined with tailored tele- or web-based feedback from healthcare practitioners, improves adherence to long-term medication [20] and antihypertensive medication.[21]"
- We have edited a sentence in the discussion to emphasise that a DI would be delivered alongside a VBI
 - o Line 563: "While the DI was considered limited in its potential to address intentional non-adherence, HCPs saw the potential for a brief face-to-face discussion (or 'VBI') with patients in addressing this gap, when delivered alongside a DI."
- We have added a few sentences in the discussion to draw attention to other potential solutions for medication adherence:
 - o Line 603: "We acknowledge the possibility that patients who are intentionally nonadherent to their medication may be unwilling to download an adherence app or receive SMS support messages. In these instances, alternative, more intensive intervention methods involving multiple BCT components may be considered appropriate, such as motivational interviewing delivered face-to-face and/or over the telephone.[33–35]"
- The authors write that "In addition, the benefit of DIs over traditional face-to-face approaches is that they do not require lengthy consultations with healthcare practitioners, whose time is scarce, especially during the current Covid pandemic." There is no reference to support this claim here (that Di can replace face to face consultation), and while I see the issue of time being scarce for health practitioners, this phrasing seems somewhat exaggerated. If an adherence conversation would indeed demand a lengthy consultation, would it then be easily

resolved by an app? And if so, would that be desirable (for whom)? I think the authors intent to make a more general claim here, either about cost-effectiveness or efficiency, please consider rephrasing

- Response: Thank you for this point. We have rephrased this as “In addition, DIs can potentially lower costs compared to traditional face-to-face approaches through reducing or eliminating consultation time required with healthcare practitioners, which may be particularly valuable at times when there is a high demand for consultations e.g. during the current Covid pandemic” (line 106)

- Discussion: Given the low recruitment/willingness to participate, I am somewhat surprised that the authors do not discuss this in their paper: f only so few people are willing to use the app when its readily available?

- Response: Whilst this is a valid point, willingness to take part in a study is not quite the same as willingness to use an app. In the paper we suggest this is where a brief face-to-face discussion with a healthcare practitioner could potentially play a role in engaging patients with a DI (please see line 514 onwards “However, a more promising way of addressing intentional non-adherence was highlighted in relation to the 5-minute VBI component prior to use of the DI. The VBI was presented as a way for HCPs to signpost patients to the DI and discuss medication taking behaviour...”)

- We have now expanded on this potential limitation in the Strengths and Limitations section:

- o Line 603: “We acknowledge the possibility that patients who are intentionally nonadherent to their medication may be unwilling to download an adherence app or receive SMS support messages.”

- Discussion, the authors write: ‘Participants felt a mobile digital intervention (DI) that provided simple medication reminders and feedback messages would help improve adherence.’ I’m not sure this follows from your results, your participants argue it would be helpful, but not that it would improve adherence (as you outline yourself, adherence is more complex than simply forgetting a dose)

- Response: Thank you for pointing this out. We have rephrased this sentence to accurately reflect the findings presented in the paper.

- o Line 552: “Participants found a mobile digital intervention (DI) that provided simple medication reminders and feedback messages acceptable.”

- Discussion: ‘and is among the first qualitative studies to gather patient views on the use of sensing technology (e.g. Wi-Fi or GPS) in a smartphone app to support medication adherence (see also Kassavou and colleagues[29]).’ There are several studies and reviews about adherence apps -also some qualitative studies- that aren’t referred to and aren’t used to contextualize and juxtapose the findings of this study. I would highly encourage the authors to

carefully review the existing evidence. eg: <https://mhealth.jmir.org/2019/1/e11919/> ,
<https://pubmed.ncbi.nlm.nih.gov/27248315/>

- Response: Thank you for your comment. We will address this in two parts, 1) the novel aspect of this study and 2) the contextualising of this study within existing evidence.

1) For this particular sentence (“and is among the first qualitative studies...”) we wanted to highlight the novel aspect of this study in relation to participants’ views on sensing technology – i.e. we collected qualitative data on the acceptability of this technology in an app targeting medication adherence specifically. We acknowledge that many studies have investigated the use of sensing technology in smartphone apps for health (e.g. please see reference 19, <https://www.sciencedirect.com/science/article/pii/S1532046417302782>)

- We identified qualitative data collected on this subject in relation to other behaviours and health conditions, such as smoking cessation and mental health (please see references 47 and 48). However, to our knowledge, there are no qualitative studies on the acceptability of sensing technology specifically for apps designed to support medication adherence.

- We have updated the sentence in the Strengths and Limitations section to clarify which part of the study presents novel data:

o Line 570: “Whilst previous research has investigated the use of sensing technology in smartphone apps for health [19], this study is among the first to gather qualitative data on the acceptability of such technology (e.g. Wi-Fi or GPS) in a smartphone app designed to support medication adherence (see also Kassavou and colleagues[29]).”

- We have also updated a sentence in the introduction to direct readers to a review of smartphone sensing technology within health apps:

o Line 124 - “Smartphone sensing technology has shown success in DIs across the domain of health and well-being (e.g. see Cornet and Holden for a review) [19] but user acceptability of such technology in a smartphone app to support medication adherence is largely unknown.”

2) Existing evidence on adherence apps:

- Thank you for your recommendations. We acknowledge that there is extensive research investigating medication adherence apps, both on effectiveness as well as usability and acceptability, some of which we reference in the introduction and discussion.

- As recommended, we have updated the discussion section to include existing evidence to help contextualise the current study (please see lines 637 – 651)

- We have also expanded a sentence in the discussion to be explicit about how the findings from this study compares with the existing literature (line 621)
 - o “This follows previous qualitative research in which mHealth interventions were deemed especially appropriate for “newbies” [34] i.e. patients with less experience in managing a health condition compared to those with established medication routines, for atrial fibrillation [35] and type 2 diabetes. [34]”

- Discussion, the authors write ‘We gained novel insights from a range of healthcare practitioners on the acceptability and feasibility of incorporating a VBI for medication adherence into a primary care consultation, a topic that has not been previously explored in-depth.’ Could the authors outline here what the new findings are?
 - Response: We have now removed the word ‘novel’ and outlined the recommendations arising from the study.
 - o Strengths and limitations section, line 582 “The recommendations arising from our findings can inform the development and implementation of a medication adherence VBI in primary care. Developers should consider the following: the importance of the practitioner-patient relationship when discussing medications, exploration of patient specific barriers to adherence, templates embedded within existing GP practice systems, and sufficient training for HCPs.”

- Discussion, the authors write ‘Our findings can inform the development and implementation of a medication adherence VBI in primary care.’ Please outline how?
 - Response: Thank you for this, we have now clarified how the findings can be used in the development of future interventions.
 - o line 582 “The recommendations arising from our findings can inform the development and implementation of a medication adherence VBI in primary care. Developers should consider the following: the importance of the practitioner-patient relationship when discussing medications, exploration of patient specific barriers to adherence, templates embedded within existing GP practice systems, and sufficient training for HCPs.”
 - We have also added detail to the ‘recommendations’ section, outlining how researchers and/or developers can use the findings.
 - o Line 695: “Our findings indicate the importance of comprehensive training for healthcare professionals which incorporate the principles of shared decision-making and the skills to deliver the intervention in under five minutes, as well as proficiency in using a DI. Lastly, the template for HCPs to complete the VBI and/or enter patients’ details into the DI should be user-friendly and embedded into existing GP practice software systems.”

Reviewer: 3

Ms. Ivona Mostarac, Sunnybrook Health Sciences Centre Comments to the Author:

Thank you for the opportunity to review your revised manuscript. I appreciate your consideration of my feedback and I feel you have addressed my concerns adequately in your revision.

Additional changes to the manuscript

- revised word count